# ITERATIVE GANS FOR ROTATING VISUAL OBJECTS

**Ysbrand Galama & Thomas Mensink**
Computer Vision Group
University of Amsterdam
`ysbrand@ygalama.nl & thomas.mensink@uva.nl`

## ABSTRACT

We are interested in learning visual representations which allow for 3D manipulations of visual objects based on a single 2D image. We cast this into an image-to-image transformation task, and propose Iterative Generative Adversarial Networks (IterGANs) to learn a visual representation that can be used for objects seen in training, but also for never seen objects. Since object manipulation requires a full understanding of the geometry and appearance of the object, our IterGANs learn an implicit 3D model and a full appearance model of the object, which are both inferred from a single (test) image. Moreover, the intermediate generated images from IterGANs can be used by additional loss functions to increase the quality of all generated images without the need for additional supervision. Experiments on rotated objects show how iterGANs help with the generation process.

## 1 INTRODUCTION

In our ongoing research we are interested in manipulating visual objects and scenes, *e.g.* rotate the object for a few degrees. Manipulating objects require an expectation about the appearance and the geometrical structure of the *unseen* part of the object. Humans clearly have such an expectation based on an understanding of the physics of the world, the continuity of objects, and previously seen (related) objects. We aim to learn an implicit 3D representations, which can be inferred from a single 2D image, using pairs of images of manipulated objects as training data.

In order to learn a representation for object manipulation, we cast this problem into an image-to-image transformation task. Where the goal is to transform an input image to an target image, following a pre-defined 3D transformation of the object inside. For now, we focus on a specific instance of this general problem: the object in the target image is $30^o$ rotation of the input image.

For this task, we propose the use of Iterative Generative Adversarial Networks (IterGANs), an extension of the image-to-image generator GANs (Isola et al., 2017). A fundamental difference between our task and the tasks explored in Isola et al. (2017) is, that when translating a map into an aerial image there exists a one-to-one pixel relation between the input and the output image. In our case, however, pixels have long range dependencies, depending on the geometry and appearance of the rotating object. IterGANs break these long dependencies into a series of shorter dependencies. Moreover, IterGANs allow to include unsupervised loss functions measuring the quality of the intermediate generated images to improve the overall transformation quality.

## 2 ITERGANS

In psychology it has been shown that the time to identify whether two rotated objects are the same, depends linearly on the degree of rotation between two objects, see Fig. 1(*right*) (Shepard & Metzler, 1971). Therefore, we propose to iteratively call the generator of the GAN network $k$ times:

$$B^k = \mathcal{G}_\theta(\mathcal{G}_\theta(\mathcal{G}_\theta(\mathcal{G}_\theta(\mathcal{G}_\theta(\mathcal{G}_\theta(A)))))) = \mathcal{G}_\theta^k(A). \tag{1}$$

Each generator now has to rotate its input image only by a few degrees, until the final rotation has been reached. The iterative nature of the IterGAN is illustrated in Fig. 1 (*left, in orange*).

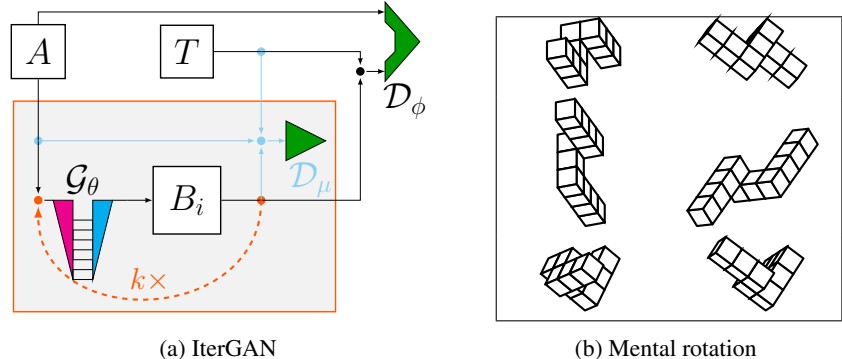



(a) IterGAN           (b) Mental rotation



Figure 1: **IterGAN** (*left*): Illustration of IterGAN network, showing the iterative nature of the generation process (*orange*) and the additional discriminator on the intermediate images (*blue*). **Mental rotation** (*right*): Research has shown there exist a linear relation between reaction time to identify matching pairs and difference in rotation between the blocks (Shepard & Metzler, 1971).
*How long does it take you to find the non-matching pair?*

Our proposed network has the same number of parameters as a network with a single generator, and for learning we can use the same generator and discriminator losses as in Isola et al. (2017):

$$\mathcal{L}_{\mathcal{G}}^{(\text{IG})} = H[\mathcal{D}_{\phi}(A, B^k), 1] + \lambda_{L_1} L_1(B^k, T), \tag{2}$$

$$\mathcal{L}_{\mathcal{D}}^{(\text{IG})} = H[\mathcal{D}_{\phi}(A, T), 1] + H[\mathcal{D}_{\phi}(A, B^k), 0], \tag{3}$$

where the cross-entropy loss ($H$) between the input image $A$ and the generated image $B^k = \mathcal{G}_{\theta}^k(A)$ is used, as well as the $L_1$ loss between the generated image $B^k$ and the target image $T$.

**Unsupervised Iterative Discriminator** An interesting property of the IterGAN model is that it allows to include additional losses on the intermediate generated images $\{B^i\}_{i=1}^{k-1}$. Here we explore adding a discriminator to enforce the implicit assumption that also intermediate generated images should look realistic.

We include a discriminator, unconditioned on the original input image to tell apart image A or T from any of the generated images $\{B^i\}_{i=1}^k$, see Fig. 1 (*left, in blue*). The generator, on the other hand, aims to fool this discriminator (as well as the main discriminator). This results in:

$$\mathcal{L}_{\mathcal{G}}^{(\text{IG+U})} = \mathcal{L}_{\mathcal{G}}^{(\text{IG})} + \lambda_u \cdot H[\mathcal{D}_{\mu}(B^i), 1] \tag{4}$$

$$\mathcal{L}_{\mathcal{D}}^{(\text{IG+U})} = \mathcal{L}_{\mathcal{D}}^{(\text{IG})} + \lambda_u \cdot \left( H[\mathcal{D}_{\mu}(B^i), 0] + H[\mathcal{D}_{\mu}(A \vee T), 1] \right) \tag{5}$$

where $B^i$ is uniformly sampled from $\{B^i\}_{i=1}^k$, either $A$ or $T$ is used, and $\lambda_u$ is an additional hyper-parameter. Since we do not require additional labeled data, we coin this the IG+U model.

**Object mask specific reconstruction loss** The dataset we use for our experiments contains (rotated) objects against a black background. In order to focus the reconstruction mostly on the objects, we use a variant of the L1-loss, which uses the provided binary mask $M$:

$$L_1^{\text{M}} = \frac{2}{|M|} \sum_{xy} M_{xy} Z_{xy} + \frac{1}{|\neg M|} \sum_{xy} \neg M_{xy} Z_{xy}, \qquad \text{with} \quad Z_{xy} = \sum_c |T_{xyc} - B_{xyc}^k|. \tag{6}$$

This loss weights the object twice as important as the background. We refer to this as IG$_{\text{M}}$ model.

## 3 EXPERIMENTS

Our (preliminary) experiments use the Amsterdam Library of Object Images (ALOI) Geusebroek et al. (2005), which contains images of 1,000 household objects, photographed from different viewing directions by rotating the objects in steps of $5^o$, resulting in 72 images per object. For training we use a set of 800 objects in 36 pairs of $30^o$ (28.8k image pairs). For testing we use 100 objects from the train set, with different start (and thus target) rotations (3.6k image pairs). We train all models for 20 epochs, with the hyper parameters from Isola et al. (2017) and (if used) $\lambda_u = 0.1$.

|            | $L_1$              | $L_1^{\mathrm{M}}$  | $D_{\mathrm{KL}}$   |
|------------|--------------------|---------------------|---------------------|
| Identity   | $.022 \pm .020$    | $.298 \pm .154$     | $0.480 \pm 0.520$   |
| Projective | $.138 \pm .032$    | $.457 \pm .156$     | $1.407 \pm 0.810$   |
| Pix2pix    | $.014 \pm .009$    | $.210 \pm .092$     | $1.329 \pm 1.333$   |
| $IG^6$     | $.014 \pm .008$    | $.200 \pm .084$     | $1.234 \pm 1.261$   |
| $IG^6_{+U}$ | $.016 \pm .010$   | $.239 \pm .099$     | $1.567 \pm 1.301$   |
| $IG^6_M$   | $.013 \pm .009$    | $.162 \pm .060$     | $1.219 \pm 1.240$   |
| $IG^6_{M+U}$ | $.012 \pm .008$  | $.147 \pm .055$     | $1.019 \pm 1.134$   |

Figure 2: **Model comparison** (*left*): Evaluation of different models (*top*) and cumulative plot of $L_1^{\mathrm{M}}$ measure (*bottom*). **Qualitative examples** (*right*): IterGANs without additional losses produce artefacts in intermediate images which disappear in final image (*top, first versus second row, zoom in for more details*), and examples of final images generated by different models (*bottom*). The proposed $IG^6_{M+U}$ model retains most sharp details while rotating, and outperforms all other models.

**Comparison to Baselines** In the first experiment, we compare different IG models, to three baselines: Pix2pix, and the identity and projective transformation. Evaluating the quality of generated images is hard by itself (Wang et al., 2002; Salimans et al., 2016), therefore we use three different measures: (i) the mean absolute distance between pixels ($L_1$ loss, also used in (Isola et al., 2017)); (ii) the object specific mask loss ($L_1^{\mathrm{M}}$ loss, Eq. 6); and (iii) the Kullback-Leibner Label divergence: $D_{\mathrm{KL}}(p(y|B^k)||p(y|T))$, to measure the similarity of the label distributions $p(y|\cdot)$, obtained from a pre-trained VGG16, between the generated and target image, inspired on Salimans et al. (2016).

From the results in Fig. 2 (*left*) we observe that for $L_1$ and $L_1^{\mathrm{M}}$ the learning methods outperform the non-learning baselines, while for $D_{\mathrm{KL}}$ the later are very strong. This is probably because VGG16 has learned to be invariant versus object viewpoint, while subtle differences in local image statistics can have a large impact. The $IG^6$ model improves the Pix2pix on $L_1^{\mathrm{M}}$ and $D_{\mathrm{KL}}$, while small, it is significant according to the non-parametric Friedman test where each image judges the two models.

**Intermediate Artefacts and Loss Functions** We observe artefacts in the intermediate images $\{B^i\}_{i=1}^{k-1}$ of the $IG^6$ model, which disappear on the final generated image $B^k$. Here, we compare qualitatively the intermediate images of IG with $k = \{3, 5, 6\}$. The results in Fig. 2 (*top right*) show that all IG models have artefacts, which we aim to suppress by adding intermediate loss functions. From the results in Fig. 2(*left*) we observe, that the $IG^6_{+U}$ model does not improve over $IG^6$ when trained using the $L_1$ loss. Probably because finding the solution satisfying both intermediate and overall quality is much harder than any solution which satisfy only final quality.

**Object Mask Objective** In the final experiment, we compare the results when using Eq. 6 in the training objective. From the results in Fig. 2(*left*) and the cumulative results in Fig. 2 (*bottom left*), we conclude that $IG^6_M$ clearly outperform the other methods, and that adding the loss functions ($IG^6_{M+U}$) further improves over $IG^6_M$. These quantitative results are supported by the qualitative examples in Fig. 2 (*bottom right*) that show more retention of details for the $IG^6_M$ model. The last row shows an object which was not seen during training.

## 4 CONCLUSION & OUTLOOK

In this paper we introduced IterGANs inspired by the mental rotation experiments[1]. In contrast to related approaches, which require either flow (Park et al., 2017), many viewpoints (Yan et al., 2016), or learn a single model per category (Zhou et al., 2016), IterGANs learn a general implicit 3D representation from pairs of real images, for rotating objects outperforming other models, and which can even be used to rotate objects never seen before. Our current work investigates how to extend IterGANs to allow for more complex 3D transformations.

---

[1]In Fig. 1b the middle objects are not only rotated, but also mirrored.

ACKNOWLEDGMENTS

This research is supported in part by the NWO VENI What&Where project. Results and code of our current research on IterGANs is available on ArXiv Galama & Mensink (2018)[2] and on GitHub[3].

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
