# OpenReview forum: "Iterative GANs for Rotating Visual Objects"
_ICLR.cc/2018/Workshop — Accept_

### Official Review · AnonReviewer2 · 2018-03-08
**Good workshop paper**

**Rating:** 7
**Confidence:** 4

**Review:**

I like the idea of training a GAN to generate rotated objects. I would not worry too much about the performance since the task itself is very challenging and interesting. This is probably one of the promising ways to connecting 3D geometry with deep learning. And the IterGAN idea is also related to some of the self-supervision works. The paper is well written.

---

> ### Public Comment · ~Thomas_Mensink1 · 2018-04-16
> **Thanks!**
>
> Thank you for the encouraging feedback. IterGANs are indeed a step towards to connecting to 3D geometry, yet one which should be explored more.

---

### Official Review · AnonReviewer1 · 2018-03-10
**Interesting approach for synthesizing object's rotation**

**Rating:** 7
**Confidence:** 4

**Review:**

The paper presents a method for simulate an object rotation by using the generator of a GAN iteratively. I like the idea of using intermediate steps to reach the final degree of rotation and the use of the additional discriminator for that process. The experiments seem adequate and the qualitative results should include an example of all intermediate steps to show how they help the rotation

---

> ### Public Comment · ~Thomas_Mensink1 · 2018-04-16
> **Thanks!**
>
> Thank you for your positive feedback.

---

### Decision · Program_Chairs · 2018-03-20
**ICLR 2018 Workshop Acceptance Decision**

**Decision:**

Accept

**Comment:**

Congratulations, your paper was accepted to the ICLR workshop.

---

> ### Public Comment · ~Thomas_Mensink1 · 2018-04-16
> **Looking forward to see you in Vancouver**
>
> Thanks